# An AI-Enhanced Strategy of Service Offloading for IoV in Mobile Edge Computing

Hongyu Peng [1,*], Xiaosong Zhang [1], Hongwu Li [2], Lexi Xu [2] and Xiaodong Wang [1]

1   Department of Artificial Intelligence, Tangshan University, Tangshan 063000, China;
    zhangxiaosong@tsc.edu.cn (X.Z.); wangxiaodong@tsc.edu.cn (X.W.)
2   Research Institute, China Unicom, Beijing 100048, China; lihw153@chinaunicom.cn (H.L.)
*   Correspondence: penghongyu@tsc.edu.cn; Tel.: +86-0315-2033957

**Abstract:** A full connected world is expected to be introduced in the sixth generation mobile network (6G). As a typical fully connected scenario, the internet of vehicle (IoV) enables intelligent vehicle operations via artificial intelligence (AI) and edge computing technologies. Thus, integrating intelligence into edge computing is, no doubt, a promising development trend. In the future of vehicular networks, a massive variety of services need powerful computing resources and higher quality of service (QoS). Existing computing resources are insufficient to match those increasing requirements. Most works on this problem focused on finding the power-delay's trade-off, ignoring QoS and stable load balance. In this study, we found that the computing power and redundancy of vehicles' in IoV is increasing. So, those redundant computing resources are possible to be used to solve the shortage of computing resource. CNN is a typical AI technique. This technology is very suitable for solving the problems in this article. So, we adopted CNN technique of AI to design and algorithm of convolutional long short-term memory (CN_LSTM) based traffic prediction (ACLBTP). ACLBTP was designed to gain the predicted number of vehicles belonging to the edge node. Secondly, according to the problem of insufficient computing resources on remote servers, we found that a large amount of redundant computing resources exist in edge nodes. So, we used edge computing technique to solve the problem of insufficient computing resources on remote servers. ASOBCL was designed to distribute computing tasks to edge nodes. Meanwhile, an intelligent service offloading framework was provided in this article. Based on the framework, an algorithm of improved gradient descent (AIGD) was created to accelerate the speed of iteration. So, the ACLBTP's convergence of convolutional neural network (CNN) based on AIGD was able to be accelerated too. In ASOBCL, a sorting technique was adopted to speed up the offloading work. Simulation results demonstrated the fact that the prediction strategy designed in this paper had high accuracy. The low offloading time and maintaining stable load balance were gained via running ASOBCL. Low offloading time means short response time. Additionally, the QoS was guaranteed. So, these strategies designed in this paper were effective and valuable.

**Keywords:** 6G; IoV; AI; edge computing; QoS; CNN; LSTM





## 1. Introduction

A full connected world is expected to be introduced in the sixth generation mobile network (6G) [1]. Sixth generation (6G) network faces the challenge of working efficiently and flexibly in a wider range of scenarios [2]. The autonomous management capability of 6G systems is enhanced to satisfy various application requirements, such as mobile broad bandwidth and low latency [3]. As a typical fully connected scenario, the internet of vehicle (IoV) was selected as the research subject in this paper. According to GE Digital, new technology of IoT is estimated to unlock manufacturing savings and benefit 46 percent of the global economy [4]. The internet of vehicles (IoV) is a communication paradigm that connects the vehicles to the Internet for transferring information between the networks [5].

The internet of vehicles, as an important subset of internet of things, developed rapidly in recent times. The development of IoV delivers new insights into application of IoT. Internet of vehicles (IoV) is constructed with a number of connected vehicle devices which provide a variety of services [6]. At the same time, the progress of IoV substantially pushes the development of intelligent transportation [7]. With the increasing growth of sensors' and perception devices' amount, more and more valuable information is able to be obtained from the surrounding environment via vehicles [8]. In the meantime, the data processing capability of on-board equipment is constantly improving with the hardware's innovation and performance upgrade [9]. Meanwhile, IoV services (e.g., auto navigation, traffic forecast and route planning, etc.) are most delay sensitive. However, the conventional computing resources are not enough to meet these real time requirements of services [10]. Advanced intelligent vehicular applications (e.g., intelligent road environment perception, intelligent decision making and vehicle behavior controlling, etc.) are envisioned in the future [11]. These intelligent vehicular applications need powerful computing capability, low latency and stable load balance [12]. According to the problem of conventional computing, resources are not enough to meet these real time requirements of services, and existing works mostly ignored the impact of service offloading and of QoS [13]. The summary of problems in existing works is given as follows.

With these observations from Table 1, we can understand clearly that it is challenging to gain high QoS and stable load balance at the same time. In this article, the redundant computing power of vehicles' in IoV was fully utilized to achieve this goal. Firstly, we achieved an intelligent computing and service offloading architecture for high QoS. Then, a series of strategies were designed for gaining low offloading time and maintaining stable load balance.

**Table 1.** Summary of Problems in Existing Works.

| Problems | Existing Works |
|---|---|
| Ignoring the problems of load balance and the computing ability of edge nodes. | Satveerrs S et al. provide a plan on service offloading to find the best power-delay's trade-off [14]. |
| Ignoring the problem of delay. At the same time, the quality of service (QoS) is not considered properly. | H. Liu et al. propose parked vehicle edge computing for distributed task execution [15,16]. |
| Ignoring the problem of large consumption of storage space. In some cases, this may lead to serious consequences. | To provide high-quality information services, Z. Su et al. suggest a strategy for caching content in parked vehicles in advance [17]. |
| Redundant computing and storage resources are not been considered for reasonable utilization. At the same time, the quality of service (QoS) is not considered properly. | W Sun et al. suggest edge computing is able to provide distributed computing service through small-scale data centers near the edge of the network [18]. |

The key contributions of this paper are summarized as follows.

(1) Develop an AI-based framework to deploy these strategies designed in this paper for IoV during the service's offloading process. In this framework, high QoS and stable load balance are gained via running these strategies. This framework has important practical significance for smart transportation especially.

(2) Adopt AIGD (Algorithm of Improved Gradient Descent) to improve the speed of iteration. So, the convergence efficiency of CNN based on AIGD is able to be improved significantly. As a result, the speed of the strategies based on AIGD is faster than normal. This means lower time complexity and lower delay.

(3) Design ACLBTP (Algorithm of CN_LSTM Based Traffic Prediction) to gain the predicted number of vehicles. These vehicles are selected to be offloaded services.

(4) Conduct ASOBCL (Algorithm of Service Offloading Based on CN_LSTM) to offload the services. A sorting technique was adopted in this algorithm. So, the work of offloading in this strategy is more efficient than normal. This means high QoS. This strategy is able to be deployed in scenarios with responsive requirements.

The rest of this paper is organized as follows. The related works of this paper is introduced in Section 2. Section 3 presents the framework of system. The model of system is proposed in Section 4. These strategies of this paper are discussed in Section 5. The simulation analysis is given in Section 6. Section 7 proposes the conclusion and future works of this paper.

## 2. Related Works

IoV gained explosive advancement due to the growth of sensors technique [19]. Because vehicles are integrated with more and more computing resources, according Moore's law, those selected IoV services are able to be offloaded to those selected vehicles rather than those edge nodes and remote services. Benefiting from those selected vehicles, the quality of service is improved and the transmission time is reduced dramatically.

Many studies examined the effectiveness for service offloading. Wan et al. designed an improved computation offloading method in a 5G environment for IoV, which could address the challenge of selecting appropriate different destination [20]. Song et al. proposed a hierarchical edge architecture with the intention of improving the 5G-based optimal mobile system [21]. In order to select suitable service and vehicles, it is necessary and imperative to design an appropriate offloading method. Ma et al. adapted LSH to design indexes of feature vectors. The purpose was to improve the searching speed [22]. Luo et al. noted that the MEC-IoV realized the extensive communication ability in the edge of network [23]. Kai et al. focused on coordinating the vehicular layer and edge layer, and they jointly utilized heterogeneous edge computing frameworks and advance IoV systems [24].

However, to the best of our knowledge, current studies about service offloading for IoV focus on service utility and privacy security, but neglect the load balance and quality of service. In this paper, an AI-enhanced strategy of service offloading was designed, and the load balance and quality of service were improved.

## 3. The Framework of System

In this section, the notations summary of this paper and the framework of this system is designed and expounded. Firstly, the notations summary of this paper is described in Table 2.

**Table 2.** Summary of Notations in Problem Formulation.

| Notations | Descriptions |
|:---:|:---:|
| $D^{1/2}$ | the non-singular matrix |
| $R$ | the Raleigh quotient |
| $v$ | the given direction |
| $H$ | the function of Hessian |
| $\alpha$ | the gradient descent |
| $\gamma$ | the eigenvalues of H |
| $\eta$ | the eigenvectors of H |
| $S_e$ | the set of edge nodes |
| $S_s$ | the set of services |
| $R_j$ | the resource utilization of j-th EN |
| $R_{ave}$ | the average resource utilizations of ENs |
| $l_b$ | the load balance |
| $\theta$ | the parameters of function |
| $\beta$ | the learning rate |
| $\mu$ | the damping factor |

It can be seen from Table 2 that the descriptions of notations are summarized. Then, the framework of system was developed as follows.

In this article, we designed the algorithm of convolutional long short-term memory (CN_LSTM) based traffic prediction (ACLBTP). The ACLBTP was deployed in the edge nodes. The computing ability and storage of edge node was limited compared with the

remote servers in this framework. So, we adopted LSTM to design ACLBTP. Meanwhile, an algorithm of improved gradient descent (AIGD) was created to accelerate the convergence speed of ACLBTP. The gradient descent method was suitable for light-weight datasets. So, the AIGD was deployed in the edge nodes too. In this proposed intelligent service offloading framework, ACLBTP and AIGD were deployed in edge nodes. ASOBCL was deployed in the remote servers. Firstly, ACLBTP was trained in the training datasets and accelerated the convergence speed. Secondly, ACLBTP predicted the number of vehicles and selected the vehicles belonging to the edge node. Finally, ASOBCL distributed computing tasks to these selected vehicles.

As shown in Figure 1, there were three layers in this framework. These layers were data perception and computing layer, edge layer and cloud layer. The data perception and computing layer were composed by those mobile nodes selected via AI strategy deployed in edge nodes. These mobile nodes were able to exchange information with each other. The information included traffic data and road conditions, etc. These mobile nodes uploaded perception data to the edge nodes in the edge layer. The strategy deployed in edge node offloaded computing task to the mobile nodes. At the same time, the resources were allocated to these mobile nodes via the strategy.

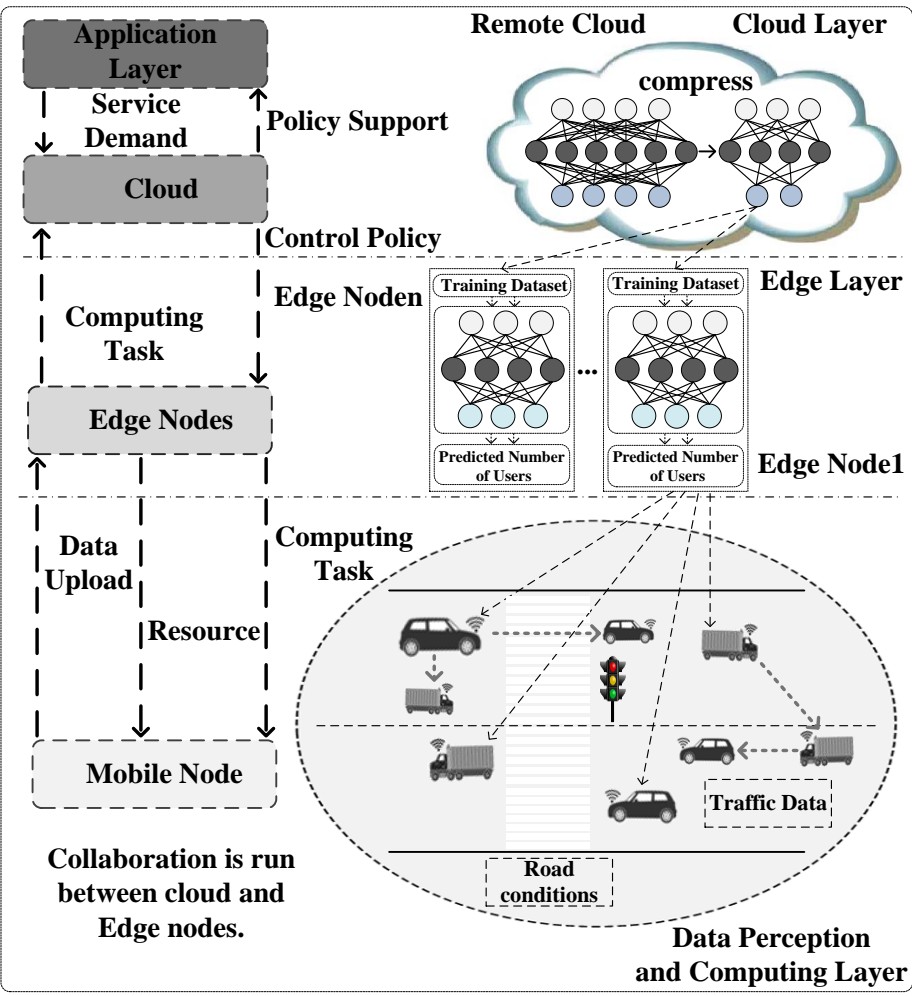

**Figure 1.** The Framework of System.

The edge layer was composed of edge nodes. The AI strategy designed in this paper was deployed in these edges. The AI strategy was run in the training set and the predicted number of mobile nodes was gained. These edge nodes assigned hard computing tasks to the cloud. The cloud sent control policy back to these edge nodes. The service demands

were provided to the applications in the application layer. At the same time, policy supports were given to these applications via cloud.

The collaboration was run between cloud and edge nodes. All the layers in the framework interacted with each other. By analyzing the datasets of services, edge knowledge bases were built. The knowledge was able to be used for the prediction of the traffic patterns. In this paper, it was used to predict the number of mobile edge nodes.

This framework was suitable for fully connected networks. In the perception layer and edge layer, each entity was able to communicate with each other. Meanwhile, there were enough redundant computing and storage resources in the vehicles of this framework. Redundant computing and storage resources in the vehicles is a prerequisite for this framework. In the future, security will be the focus of our next move. Suitable encryption techniques will be gained to ensure information and privacy security.

In this part, we analyze the complexity of the framework. Assuming that the framework of this paper is represented with F, the complexity of this framework is mainly determined by these edge nodes and connections between these edge nodes. We mapped the framework into a graph. The graph is represented by G.

$$G = (V,E) \tag{1}$$

where V is the set of edge nodes. E is the set of connections between edge nodes.

The service in $E_i$ is represented by $S_i$.

$$S_i = \{F,C,J_i\} \tag{2}$$

where F is the function of service. C is the computing time. $J_i$ represents the number of implementations of service $S_i$. Assuming the maximum number of services in G is N. In this paper, complexity of this framework was considered in the worst-case scenario. This complexity of this frame is represented as follows:

$$O(F) = N \times (J_i)^2 \tag{3}$$

From (3), it can be seen that complexity of this proposed framework was moderate. The complexity of this framework was mainly determined by the number of edge nodes. The fewer edge nodes, the lower the complexity of the framework. The reduction in vertices in G also meant the reduction in edges, the number of services deployed in edge nodes decreased and the connections between services decreased accordingly. In future research, we will investigate how to reduce the number of edge nodes by eliminating non-schedulable paths while ensuring a certain level of QoS. The low complexity of the framework was able to be gained via reducing the number of nodes.

The models of system based on the framework above are designed in the next section.

## 4. The Model of System

Precondition was designed via introducing a linear change of variables.

$$\alpha = D^{\frac{1}{2}}\theta \qquad \theta \in R \tag{4}$$

where $D^{\frac{1}{2}}$ is a non-singular matrix. Then, a new function is able to be designed as

$$f(\theta) = f(D^{-\frac{1}{2}}\alpha) \tag{5}$$

The gradient of the function is expressed as follows.

$$f'(\alpha) = D^{-\frac{1}{2}}f'(\theta) \tag{6}$$

The H in this paper is defined as

$$H = f''(\theta) \tag{7}$$

The Hessian of the function is expressed as follows.

$$f''(\alpha) = \left(D^{-\frac{1}{2}}\right)^{T} H D^{-\frac{1}{2}} \tag{8}$$

A gradient descent iteration for the transformed function is able to be gained from (6) and (8).

$$\alpha_t = \alpha_{t-1} - \lambda\, f'(\alpha) \tag{9}$$

According to the relationship of $\theta$ and $\alpha$, A gradient descent iteration for $\theta$ is able to be obtained as follows.

$$\theta_t = \theta_{t-1} - \lambda D^{-1} f'(\theta) \tag{10}$$

The Raleigh quotient R is defined as

$$R(H, v) = \frac{v^{T}(Hv)}{v^{T}v} \tag{11}$$

where v is a given direction. R is used to measure the amount of curvature.

The Raleigh quotient R is able to be decomposed as follows.

$$R(H, v) = \sum_{i}^{n} \gamma_i \eta_i \eta_i^{T} v \tag{12}$$

where $\gamma_i$ is the eigenvalues of H. $\eta_i$ is the eigenvectors of H. So, Hv is able to be gained from (11) and (12).

The set of EN (Edge Nodes) is defined as follows.

$$S_e = (e_1, e_2, ...e_n) \tag{13}$$

The set of services is defined as follows:

$$S_s = (s_1, s_2, ...s_m) \tag{14}$$

A variable $k_{i,j}$ is given to represent whether the n-th service is executed by m-th EN

$$k_{i,j} = \begin{cases} 1, & \text{i-th sercice is processed by the j-th EN} \\ 0, & \text{otherwise} \end{cases} \tag{15}$$

The resource utilization of j-th EN is calculated based on k n,m and it is given by

$$R_j = \sum_{i=1}^{m} k_{i,j} \tag{16}$$

Based on (16), the average resource utilizations of ENs $R_{ave}$ are calculated as

$$R_{ave} = \frac{1}{n}\sum_{j=1}^{n} R_j \tag{17}$$

Based on (17), the load balance $l_b$ is calculated by

$$l_b = \frac{1}{n}\sum_{j=1}^{n} \left(R_j - R_{ave}\right)^2 \tag{18}$$

The problem of designing an effective offloading method is expressed as follows.

$$\min l_b \tag{19}$$

Then, these strategies based on the model above are designed as follows.

## 5. Strategy Design

In this section, we designed three strategies. These strategies were AIGD, ACLBTP and ASOBCL. Firstly, the strategy of AIGD is described as follows.

It can be seen from Algorithm 1 that $\beta$ is the learning rate. The $\mu$ represents the damping factor. The contributions of these directions will take a large step in each direction. This will improve the speed of iteration. So, the convergence speed of CNN was improved significantly.

---

**Algorithm 1:** Algorithm of Improved Gradient Descent

---

1: Initialization $\beta$, $\mu$
2: Initialization H to 0 matrix
3: Gain the min value of $f(\theta)$
4: foreach i in (k,K)
5:   get v randomly from N(0,1)
6:   $D = D + (Hv)^2$
7:   $\theta = \theta - \beta \frac{f'(\theta)}{\sqrt{D/k} + \mu}$
8: endfor

---

Based on the strategy above, the ACLBTP is designed as follows.

It can be seen from Figure 2 that the CN_LSTM_Network Model of ACLBTP was composed of two convolution layers, a LSTM layer and a full connection layer. The input data were the traffic datasets. The output data from these convolution layers were put into the LSTM layer for extracting time features. The LSTM layer was composed of several LSTM blocks. The output from the full connection layer was the number of vehicles in each area and in every time slot. The mean squared error (MSE) was adopted as the loss function of this paper [25]. MSE was used to minimize the number of network errors.

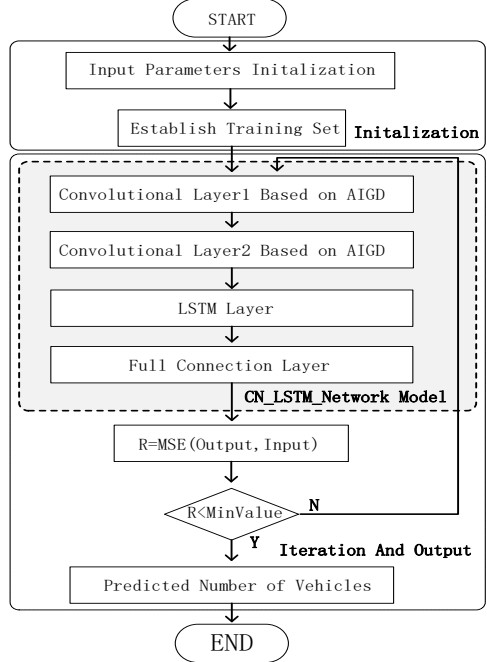

**Figure 2.** FlowChart for ACLBTP.

Base on the flowchart of strategy above, the algorithm's pseudo code was provided as follows.

It can be seen from Algorithm 2 that parameters were initialized and a training set was established firstly. Then, the convolution iteration started. The number of iteration was set to 2000 in this paper. In every iteration, the input data set was put into these convolution layers based on AIGD and the spatial feature of time series traffic data were extracted. The output data from two convolution layers were put into LSTM Layer for extracting time feature. The output data from LSTM were put into a full connection layer and the output was the number of vehicles in each area in every time slot. The iteration was continued until the number of network errors was less than the confidence interval MinValue. At last, the predicted number of vehicles was gained.

---

**Algorithm 2:** Algorithm of CN_LSTM Based Traffic Prediction

---

1: Initialization Vehicles Data Matrix: M, Layers: 3, Number of iteration: 2000, time slot:16, Number of data set for each slot:32 Number of network errors: R
2: Establish Training Set: Input Data Set, Output Result Set
3: Set Iteration Number n, Set Confidence Interval: MinValue
4: While (R< MinValue)
5:   {
6:     for i = 1 to n
7:       {
8:         Extract Spatial Feature of Time Series Traffic Data via
            two Convolution Layers Based on AIGD.
9:         The Output Data from two Convolution Layers are
            put into LSTM Layer for extracting time feature.
10:        The Output Data from LSTM are put into a full
            connection layer and the output is the number of
            Vehicles in each area in each time slot.
11:        i++
12:      }
13:    R = MSE(Output Data from LSTM, Input Data Set)
14:   }
15: Output Predicted Number of Users
16: End of Strategy

---

Based on the strategy ACLBPT above, the strategy of ASOBCL was designed as follows.

It can be seen from Figure 3 that those vehicles gained via ACLBTP were regarded as the input data of the workflow. The n was defined as the number of these vehicles. The Ss was defined as the set of services. Firstly, these vehicles were sorted via (18) by load in descending order. The order of sort was to gain the vehicle whose load was min more quickly. One service was selected from Ss and offloaded to the vehicle whose load was min. The service which was offloaded was removed from the Ss. The work of offloading continued until the Ss was empty. The sorting technique in ASOBCL ensured each offload to the vehicle with the smallest load. This guaranteed the success rate of offloading work. High success rate of offloading work gained high speed offloading work.

Based on the strategy flowchart above, the algorithm's pseudo code was provided as follows.

It can be seen from Algorithm 3 that parameters were initialized firstly. The n was initialized as the output predicted number of vehicles. The Ss was initialized as the service set. The load of each vehicle was calculated via (18). Firstly, these vehicles were sorted in descending order by the size of load. One service was selected from Ss and offloaded to the vehicle with min load. Then, the service was removed from Ss. Those vehicles were sorted in descending by load again. These steps were repeated until the Ss was empty. The strategy came to an end.

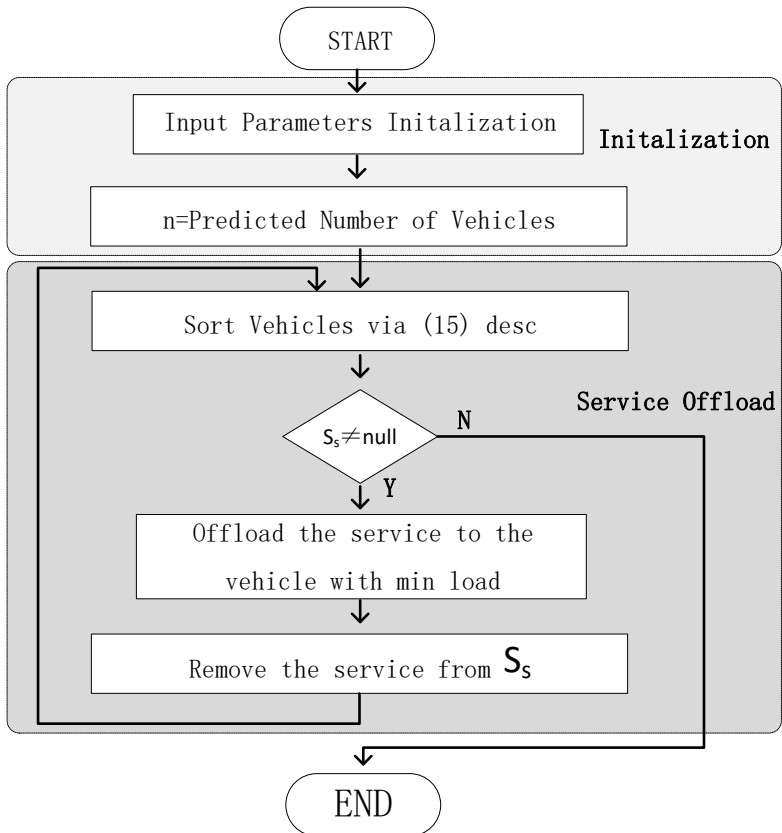

**Figure 3.** Flowchart for ASOBCL.

| **Algorithm 3:** Algorithm of Service Offloading Based on CN_LSTM Traffic Prediction |
| --- |
| 1: Initialization Output Predicted Number of Vehicles: n, Service Set: $S_s$ |
| 2: Sort n vehicles via load calculated with (18) desc. |
| 3: While($S_s$ ! = null) |
| 4:  { |
| 5:      Offload the service the vehicle with min load. |
| 6:      Remove the service from the $S_s$. |
| 7:      Sort n vehicles via load calculated with (18) |
| desc again |
| 8:  } |
| 9: End of Strategy |

The simulation analysis based on these strategies are given in the next part.

## 6. Simulation Analysis

In this paper, the technique of lightweight machine learning was adopted on the edge nodes. Pre-trained dataset was used as the input to the new machine learning task. The adoption of pre-trained dataset was able to decrease the computing complexity significantly. The pre-trained AlexNet CNN was used in MATLAB. The ILSVRC 2012 was adopted in this paper. The strategy of ASOBCL was implemented in Tensorflow. According to [26], the channel was selected with parameter of 1. The noise power density was adopted as −120 dBm.

The performance of these strategy designed in this paper was evaluated from three aspects: the prediction accuracy, the load balance and the offloading time.

### 6.1. Prediction Accuracy

We trained the ACLBTP for 2000 epochs, using a $3 \times 3$ convolution kernel. The values of the learning rate, the decay rate and the dropout rate were set as 0.15, 0.85 and 0.6, respectively. The batch size was 8. The CabSpotting dataset [27] was taken as the vehicle trajectory data set. After pre-processing, the vehicle trajectory data set was converted into a traffic data set containing 3000 pieces of data. A total of 50 percent of the available data were used for training. The remaining 50 percent of the data were taken as validation data sets.

We compared our ACLBTP model with the two existing prediction models: LSTM and Conv_LSTM [28]. The comparisons of real values and the predicted values generated by the three models are shown in Figures 4–6.

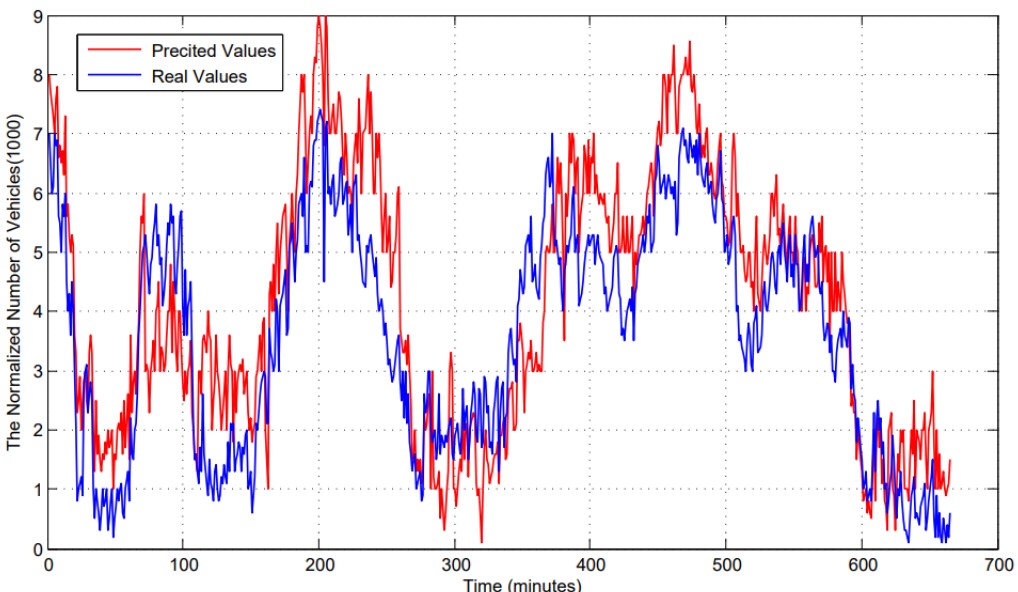

**Figure 4.** Comparisons of real values and predicted values from LSTM.

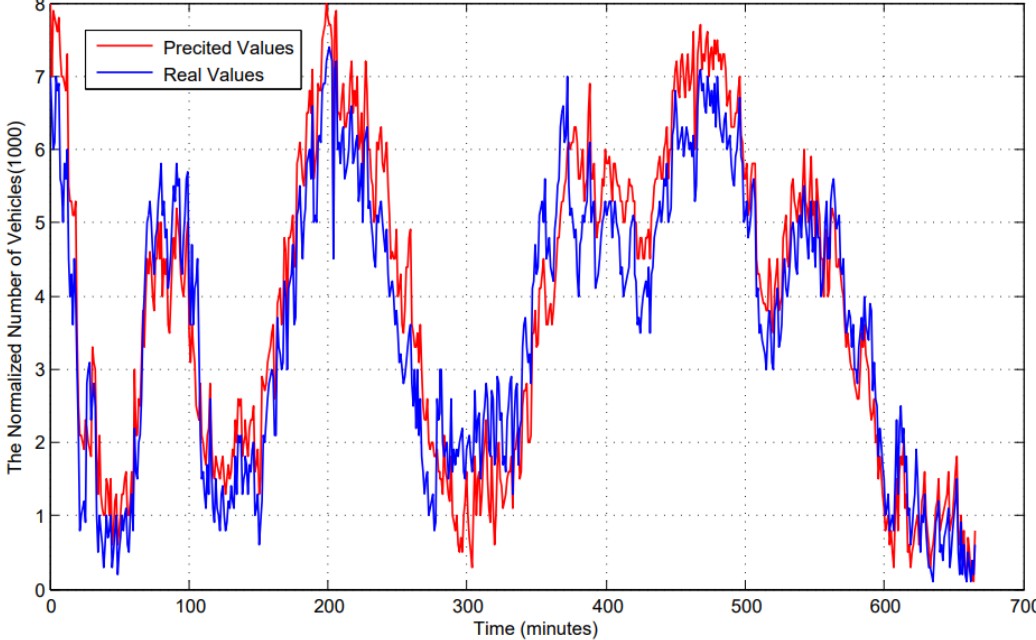

**Figure 5.** Comparisons of real values and predicted values from Conv_LSTM.

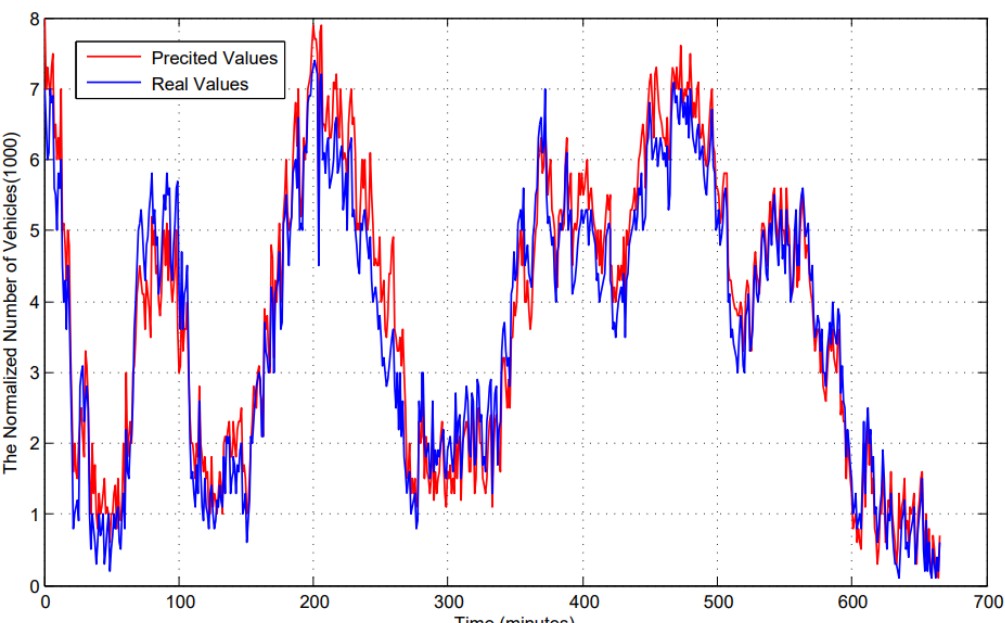

**Figure 6.** Comparisons of real values and predicted values from ACLBTP.

Figure 4 depicts the accuracy of prediction of LSTM over the varying training set size. As shown in Figure 4, the horizontal axis indicated the lapse of time. The unit was 60 s. The vertical axis represents the normalized number of mobile edge nodes. The blue line indicates the changing trends of real values. The red line indicates the changing trends of predicted values. When the time lapse was at around 100 min, the predicted value deviated significantly from the real value.

Figure 5 depicts the accuracy of prediction of Conv_LSTM over the varying training set size. As shown in Figure 5, the horizontal axis indicates the lapse of time, and the unit was 60 s. The vertical axis represents the normalized number of mobile nodes. The blue line indicates the changing trends of real values. The red line indicates the changing trends of predicted values.

Figure 6 depicts the accuracy of prediction of ACLBTP over the varying training set size. As shown in Figure 6, the horizontal axis indicates the lapse of time, and the unit was 60 s. The vertical axis represents the normalized number of mobile nodes. The blue line indicates the changing trends of real values. The red line indicates the changing trends of predicted values.

Upon comparing Figures 4 and 5 with Figure 6, it can be seen that the predicted values from ACLBTP designed in this paper were much closer to the real values than those from Conv_LSTM and LSTM. The reason was that ACLBTP can extract time and spatial features of traffic data with the help of AIGD.

### 6.2. Load Balance

In this section, comparative algorithms and results analysis are proposed. Firstly, comparative algorithms are introduced as follows.

### 6.2.1. Comparative Algorithms

There were four comparative algorithms in this paper: Service Offloading Method (SOME), First Come First Service (FCFS), Next Come First Service (NCFS) and Benchmark First Service (BCFS).

SOME: The strategy is an offloading strategy which adopts the locality-sensitive-hash (LSH) technique. This strategy is designed to offload services and promote IoV service utility and edge utility, ensuring privacy security at the same time [29].

FCFS: When the service comes, the vehicle is selected randomly to be provided to the service in turn until the set of vehicles is empty.

NCFS: When the first service comes, the vehicle is selected randomly to be provided to the service. When the following service comes, the first selected vehicle is excluded. The vehicle is selected randomly in the remaining vehicles set to be provided to the service until the set of vehicles is empty.

BCFS: According to the vehicles' computing capability, these vehicles are sorted in descending order. The services are assigned to these vehicles in turn [30].

### 6.2.2. Results Analysis

In terms of load balance, simulation verification was carried out from the perspective of the number of services in this paper. Additionally, the simulation verification was introduced from the perspective of the load balance and the services' number.

Maintaining a stable load balance was an important goal in this paper. As shown in Figure 7, the horizontal axis indicated the number of services, and the unit was 1000. The vertical axis represents the load balance. As the number of services increased, the load balances of FCFS, BCSF, NCFS, SOME and ASOBCL rose too. Among the four algorithms, the effect of ASOBCL was always the best, SOME was second, NCFS was the worst, and FCFS's effect was better than NCFS's but worse than SOME and BCFS's. FCFS and NCFS selected vehicles randomly. This led to high load balance. ASOBCL always maintained stable load balance rate in these vehicles because vehicles were selected reasonably and scientifically.

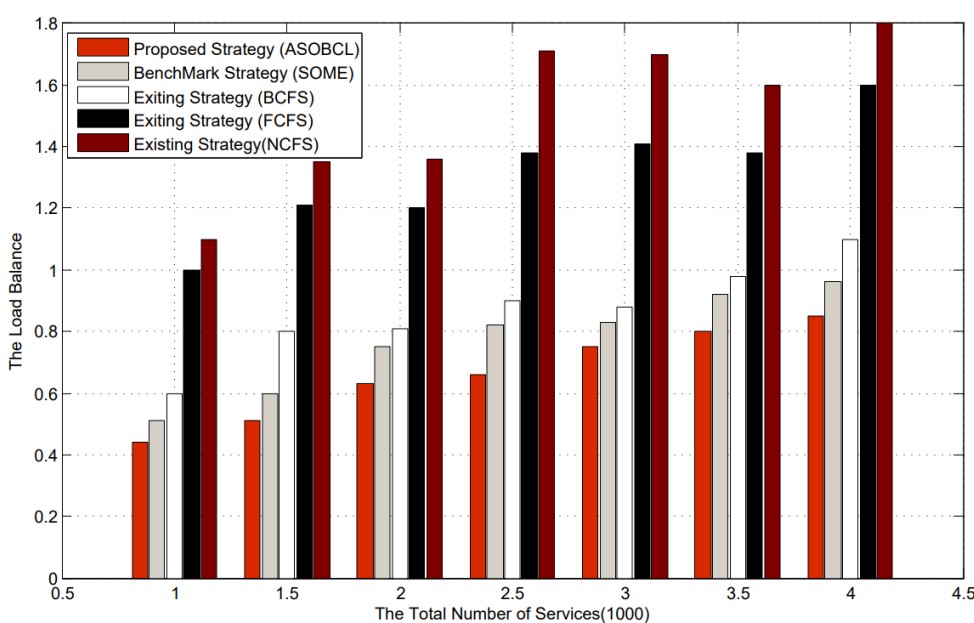

**Figure 7.** The number of services and load balance.

### 6.3. Offloading Time

In this section, comparative algorithms are adopted from the above section. In terms of offloading time, simulation verification was carried out from the perspective of the number of services in this paper. Simulation verification was introduced from the perspective of the offloading time and the services' number.

The offloading time was compared among FCFS, BCSF, NCFS, SOME and ASOBCL. As shown in Figure 8, the horizontal axis indicates the number of services, and the unit was 1000. The vertical axis represents the offloading time, and the unit was 1000 ms. As the number of services increased, the offloading time for FCFS, BCSF, NCFS, SOME and ASOBCL rose too. Among the four algorithms, the effect of ASOBCL was always the best, SOME was second, FCFS was the worst, and NCFS's effect was better than FCFS's but

worse than SOME and BCFS's. The more the number of visits, the better the offloading time of ASOBCL. Compared with the other offloading methods, ASOBCL gained lower offloading time.

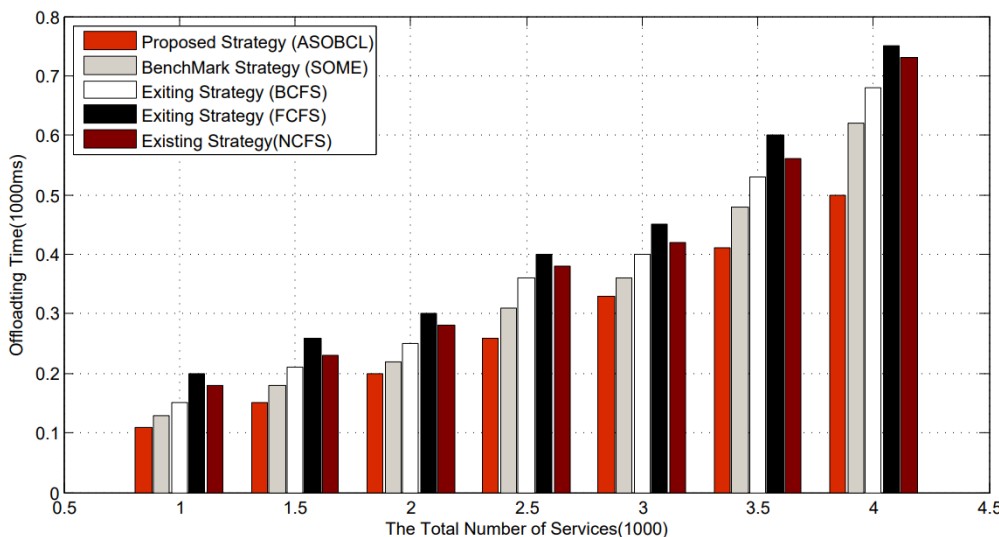

**Figure 8.** The number of services and Offloading Time.

## 7. Conclusions and Future Work

In this paper, we introduced AI in future vehicular networks, where vehicles which have enough computing power were taken as auxiliary edge nodes to provide extensive resources. Firstly, an intelligent service offloading framework was provided. Based on the framework, three strategies were proposed. The AIGD was created to accelerate the convergence of CNN. The ACLBTP was designed to gain the predicted number of vehicles belonging to the edge node. The ASOBCL was conducted to offload these services to the vehicles belonging to the edge node. The simulation results demonstrated that the prediction strategy designed in this paper had higher prediction accuracy compared with other prediction strategies. The low offloading time and maintaining stable load balance were gained via running ASOBCL. So, the effectiveness and efficiency of ASOBCL was verified by experiment evaluation. These strategies proposed in this paper will make full use of the increasing and additional computing power in vehicles in the perception and computing layer. Computing load of remote servers and general edge nodes was able to be decreased dramatically. At the same time, transmission bandwidth will be saved. Computing the load's decrease and saving of transmission bandwidth means low energy consumption. Low energy consumption leads to lower coal demand. So, a reduction in greenhouse gas emissions is gained and the greenhouse effect will be suppressed effectively. In our future work, we will devote to applying ASOBCL to real life, taking more real details of an IoV environment into account. These real details include real-time measurement of vehicles' computing power, the vehicles' speed and redundant computing power. Adopting multi-objective optimization techniques, these strategies of this paper will gain better practical significance via considering these factors fully.

**Author Contributions:** Conceptualization, H.P. and X.Z.; formal analysis, H.P.; funding acquisition, H.P.; investigation, H.P. and X.W.; methodology, H.P. and L.X.; software, H.P.; supervision, H.P.; validation, H.P. and H.L.; visualization, X.W.; writing—original draft, H.P.; writing—review and editing, L.X. All authors have read and agreed to the published version of the manuscript.

**Funding:** The Southwest JiaoTong University Cooperative Intelligent Water Project. funding number: 1200305; the Doctor Innovation Fund project and Tangshan Indoor Positioning Key Laboratory Construction project. funding number: 1401801.

**Data Availability Statement:** The data can be shared up on request.

**Conflicts of Interest:** The authors declare no conflict of interest.

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
