# Peer review of "An AI-Enhanced Strategy of Service Offloading for IoV in Mobile Edge Computing"

_electronics, doi:10.3390/electronics12122719_

Round 1
Reviewer 1 Report
The paper focuses on the 6th generation mobile network (6G), discussing the future of vehicular networks and the requirements for high-quality service (QoS) and significant computing resources. Specifically, it addresses the problem of insufficient existing resources to meet these needs by introducing an intelligent service offloading framework and associated algorithms. These include the Algorithm of Improved Gradient Descent (AIGD), the convolutional long short-term memory (CN_LSTM) Based Traffic Prediction (ACLBTP), and the Algorithm of Service Offloading Based on CN_LSTM (ASOBCL).
The paper aims to accelerate the convergence of the convolutional neural network (CNN) with AIGD, predict the number of vehicles related to the edge node using ACLBTP, and offload services to these vehicles using ASOBCL. The final algorithm implements a sorting technique to expedite the offloading process. The proposed methodologies are presented succinctly, and the simulation results claim high accuracy, short response time, and maintained stable load balance, thus promising an enhanced QoS.
The paper appears to make several significant contributions:
- It proposes an original framework for service offloading in vehicular networks, which is critical given the increasing demands on these networks and the move towards more intelligent operations.
- It offers the AIGD, ACLBTP, and ASOBCL algorithms to resolve the problems inherent in the emerging scenario.
- Using an improved gradient descent algorithm for accelerating the convergence of CNNs is quite innovative.
- The paper's use of CN_LSTM for traffic prediction and service offloading is a novel application of this neural network architecture, extending its utility beyond traditional uses.
Despite its potential, the paper has a few areas that could be improved:
- While the paper mentions AI and edge computing technologies, it doesn't delve into their roles in the proposed framework or the algorithms.
- The paper seems to lack clarity in several areas, particularly regarding the precise workings and impact of the proposed algorithms.
- It's unclear how the sorting technique in ASOBCL speeds up offloading work. Further detail or supporting references would help validate this claim.
- The simulation results demonstrate high accuracy and low offloading time, but the paper does not present the exact comparison with other techniques and details of the simulation setup. Including these would make the claims more convincing.
For improvement, the authors should provide a more detailed account of the methodologies, the roles of AI and edge computing technologies, the specifics of the sorting technique, and a comprehensive discussion of the simulation results, ideally in the context of existing methods. More detail could allow readers to appreciate the novelty and potential impact of the research fully.
In conclusion, this paper proposes an intriguing and potentially impactful framework for service offloading in vehicular networks in the 6G era. It offers novel algorithms to accelerate the convergence of CNNs and to predict and offload traffic. Despite some weaknesses in clarity and detail, the paper provides valuable insights for future research and practical applications in the field. However, thoroughly reading the full paper must provide a comprehensive evaluation.
Author Response
We would like to thank you for this valuable suggestion.
Please see the attachment

Reviewer 2 Report
Abstract: The abstract provides a brief overview of the paper's topic, focusing on the challenges faced in the Internet of Vehicles (IoV) and the proposed intelligent service offloading framework. However, it lacks clarity and conciseness. The abstract should be revised to clearly state the problem, objectives, methodology, and key findings of the study.
The introduction provides a general background on the 6th generation mobile network (6G) and the Internet of Vehicles (IoV). However, it lacks proper citation and references to support the statements made. Additionally, the introduction should provide a more comprehensive literature review to highlight the existing research gaps and the significance of the proposed strategies. The research objectives and contributions should be clearly stated.
The conclusion summarizes the proposed intelligent service offloading framework and the three strategies. However, it lacks a comprehensive discussion of the results and their implications. The future work section is vague and lacks specific details on how the proposed strategies will be applied in real-life scenarios. It would be beneficial to provide more concrete plans and considerations for future research.
1. The abstract should be revised to provide a clear and concise overview of the problem, objectives, methodology, and key findings of the study.
2. The introduction should include proper citations and references to support the statements made about the 6th generation mobile network (6G) and the Internet of Vehicles (IoV).
3. The introduction should provide a more comprehensive literature review to highlight existing research gaps and establish the significance of the proposed strategies.
4. The research objectives and contributions should be clearly stated in the introduction to provide a focused direction for the study.
5. The conclusion should include a more comprehensive discussion of the results and their implications, relating them back to the research objectives and contributions.
6. The future work section should provide more specific details on how the proposed strategies will be applied in real-life scenarios, taking into account the practical considerations and challenges.
7. The manuscript would benefit from a clear research problem statement, research questions, and hypotheses to provide a solid foundation for the study.
8. The methodology of the proposed intelligent service offloading framework should be described in detail, including the algorithms used and their underlying principles.
9. The simulation results should be presented and analyzed in a more detailed manner, providing relevant metrics and comparisons to evaluate the effectiveness and efficiency of the proposed strategies.
10. The manuscript should undergo a thorough proofreading to address grammar and language issues, ensuring clarity and coherence throughout the document.
Author Response
We would like to thank you for this valuable suggestion.
Please see the attachment.

Reviewer 3 Report
I had the pleasure of reviewing the paper “An AI Enhanced Strategy of Service Offloading for IoV in Mobile Edge 2 Computing” (electronics-2432644) submitted to Electronics. The topic is interesting and necessary, and the authors have done a good job in conducting and writing up their study but the manuscript needs to be revised taking into account the following guidelines:
1/ The introduction of the manuscript should be enriched with the literature. As this is the only theoretical section in the manuscript a better anchoring of the study to the existing scientific output is needed.
2 / The research gap needs to be better described and supported by existing literature. The authors’ arguments for studying what they do, primarily rest on gap-filling. There is little in the way of a convincing explanation as to why addressing such a gap is essential, and why the analysis of these topics merits attention within their work.
3/ The theoretical contribution should be clearly indicated. In my opinion, the manuscript also has some practical implications that should also be pointed out there.
4/ Unfortunately, the limitations of the study were not included in the Conclusion section, though they should be because they certainly exist.
5/The directions of future research must be described in more detail.
6/ Practical and societal implications/impact must be indicated in the Conclusion section.
7/ Additional linguistic proofreading of the manuscript is necessary.
Additional linguistic proofreading of the manuscript is necessary.
Author Response

(The authors gave the same response as above.)

Reviewer 4 Report
An AI Enhanced Strategy of Service Offloading for IoV in Mobile Edge Computing
Few specific facts about the manuscript are given below:
1. From line number 45 to 53, problems of existing work are discussed; it should be in a tabular comparison format, then compare your work with existing work.
2. There should be a separate section for related works.
3. For large datasets gradient descent method is not appropriate. How this method is appropriate for the proposed framework?
4. ACLBTP is using CN with LSTM. Why specifically is this combination used while there are more efficient combinations available?
5. LSTM is slow on large datasets; how is this problem mitigated in the proposed framework?
5. Please include a complexity analysis of the proposed framework.
6. Please include the specific limitations of the proposed framework and future work.
7. Include proper mapping and how the proposed algorithms are utilized as per the specifications.
8. Revise the conclusion section.
Minor editing of the English language is required; some sentences need revisions.
Author Response

(The authors gave the same response as above.)

Round 2
Reviewer 2 Report
Based on the revisions made by the author, all the comments and suggestions have been successfully incorporated, strengthening the manuscript. I recommend its publication.
Author Response
We would like to thank you again for your valuable suggestions.
Please see the attachment.

Reviewer 3 Report
Dear Authors,
Thank you for the opportunity to review the revised version of the manuscript ‘An AI Enhanced Strategy of Service Offloading for IoV in Mobile Edge Computing’ submitted to Electronics (Manuscript ID: electronics-2432644). In the current version, the manuscript is more consistent and complete. First of all, I appreciate the changes in the introduction consisting in clarifying the research gap and indicating a better justification of the research problem. The authors improved the theoretical input, practical implications, limitations, and future research directions. Pointing out future research directions that were actually missing in the previous version of the paper also significantly increases the value of the manuscript.
Detailed modifications that have been introduced in the current version of the manuscript concern the following issues:
1/ The introduction has been enriched with the literature. A separate section for related works has been added to the article.
2/ The research gap has been better described and supported by existing literature.
3/ The theoretical contribution has been clearly indicated. Some practical implications have been pointed out too.
4/ The limitations of the study have been included in the Conclusion section.
5/ Managerial and societal implications and future research directions have been corrected and enriched.
6/ The manuscript has been linguistically corrected.
Considering the above, I believe that the manuscript can be published in its present form.
Author Response

(The authors gave the same response as above.)

Reviewer 4 Report
An AI Enhanced Strategy of Service Offloading for IoV in Mobile Edge Computing
Authors have addressed all the concerns except complexity analysis. The complexity analysis should be analyzed as per the standard mechanism with notations rather than general discussions for better understanding to the readers.
In few places, minor editing of english language is required, so thorough proofreading is needed before publishing.
Author Response

(The authors gave the same response as above.)
